# Relationship of Daily Coffee Intake with Vascular Function in Patients with Hypertension

**DOI:** 10.3390/nu14132719

**Published:** 2022-06-29

**Authors:** Takayuki Yamaji, Takahiro Harada, Yu Hashimoto, Yukiko Nakano, Masato Kajikawa, Kenichi Yoshimura, Chikara Goto, Aya Mizobuchi, Shunsuke Tanigawa, Farina Mohamad Yusoff, Shinji Kishimoto, Tatsuya Maruhashi, Ayumu Nakashima, Yukihito Higashi

**Affiliations:** 1Department of Cardiovascular Medicine, Graduate School of Biomedical Sciences, Hiroshima University, Hiroshima 734-8551, Japan; ts5216yt@gmail.com (T.Y.); harataka0513@gmail.com (T.H.); teriadeshio@yahoo.co.jp (Y.H.); ynakano@xj8.so-net.ne.jp (Y.N.); 2Division of Regeneration and Medicine, Medical Center for Translational and Clinical Research, Hiroshima University Hospital, Hiroshima 734-8551, Japan; m-kajikawa@hiroshima-u.ac.jp (M.K.); keyoshim@hiroshima-u.ac.jp (K.Y.); 3Department of Biostatistics, Medical Center for Translational and Clinical Research, Hiroshima University Hospital, Hiroshima 734-8551, Japan; 4Department of Rehabilitation, Faculty of General Rehabilitation, Hiroshima International University, Hiroshima 739-2695, Japan; t-goto@hs.hirokoku-u.ac.jp; 5Department of Regenerative Medicine, Research Institute for Radiation Biology and Medicine, Hiroshima University, Hiroshima 734-8551, Japan; mizobuchiaya3@gmail.com (A.M.); tanishi@hiroshima-u.ac.jp (S.T.); drfarinamyusoff@gmail.com (F.M.Y.); shinji0922k@yahoo.co.jp (S.K.); 55maruchin@gmail.com (T.M.); 6Department of Stem Cell Biology and Medicine, Graduate School of Biomedical Sciences, Hiroshima University, Hiroshima 734-8551, Japan; ayumu@hiroshima-u.ac.jp

**Keywords:** coffee, endothelial function, flow-mediated vasodilation, nitroglycerine-induced vasodilation

## Abstract

We evaluated the relationship of daily coffee intake with endothelial function assessed by flow-mediated vasodilation and vascular smooth muscle function assessed by nitroglycerine-induced vasodilation in patients with hypertension. A total of 462 patients with hypertension were enrolled in this cross-sectional study. First, we divided the subjects into two groups based on information on daily coffee intake: no coffee group and coffee group. The median coffee intake was two cups per day in the coffee group. There were significant differences in both flow-mediated vasodilation (2.6 ± 2.8% in the no coffee group vs. 3.3 ± 2.9% in the coffee group, *p* = 0.04) and nitroglycerine-induced vasodilation (9.6 ± 5.5% in the no coffee group vs. 11.3 ± 5.4% in the coffee group, *p* = 0.02) between the two groups. After adjustment for confounding factors, the odds ratio for endothelial dysfunction (OR: 0.55, 95% CI: 0.32–0.95) and the odds ratio for vascular smooth muscle dysfunction (OR: 0.50, 95% CI: 0.28–0.89) were significantly lower in the coffee group than in the no coffee group. Next, we assessed the relationship of the amount of daily coffee intake with vascular function. Cubic spline curves revealed that patients with hypertension who drank half a cup to 2.5 cups of coffee per day had lower odds ratios for endothelial dysfunction assessed by flow-mediated vasodilation and vascular smooth muscle dysfunction assessed by nitroglycerine-induced vasodilation. Appropriate daily coffee intake might have beneficial effects on endothelial function and vascular smooth muscle function in patients with hypertension.

## 1. Introduction

Coffee is one of the most popular beverages worldwide. Alcohol and tobacco are considered to be harmful for health [1,2], whereas coffee is considered to be beneficial for health [3]. The World Health Organization (WHO) issued a warning that alcohol consumption causes about three million deaths and that tobacco causes more than eight million deaths worldwide every year [4,5]. On the other hand, several lines of evidence have shown that individuals who drink coffee have a lower risk for all-cause mortality [6]. Previously, coffee had been considered to be carcinogenic [7]. However, in 2016, the International Agency for Research for Cancer and WHO reported that there is no evidence that coffee increases the risk of cancer [8]. On the other hand, the relationship between the hazard ratio for cardiovascular disease (CVD) and daily coffee intake is still controversial. Some studies have shown that coffee intake or even excessive intake can be associated with decreased CVD events [9,10,11,12]. It is also still not clear whether coffee intake has beneficial effects or not in patients with hypertension. A meta-analysis revealed that light to moderate coffee intake (1 to 3 cups/day) increased the risk of hypertension [13]. On the other hand, another meta-analysis showed that regular coffee intake has no effects on blood pressure level and risk of hypertension [14]. A previous study showed that high coffee intake (≥4 cups/day) increases the risk of stroke [15]. However, other studies have shown that even high coffee intake has no effects or beneficial effects on CVD events [12,16]. It is also still not clear whether daily coffee intake increases the risk for CVD events in patients with hypertension. 

Endothelium cells maintain endothelial function by secreting several vasodilators such as nitric oxide (NO) and endothelium-derived hyperpolarizing factor [17,18]. Endothelial function is impaired by cardiovascular risk factors in the initial step of atherosclerosis [19,20,21,22], and vascular smooth muscle function is impaired in the progressed state of atherosclerosis [23]. Measurement of flow-mediated vasodilation (FMD) in the brachial artery is the most popular tool for assessing endothelial function and measurement of nitroglycerine-induced vasodilation (NID) is the most popular tool for assessing vascular smooth muscle function [18,23]. Both FMD and NID are associated with cardiovascular risk factors, are targets for the treatment of cardiovascular disease, and are predictors for cardiovascular events. Although previous studies have shown that FMD and NID are correlated with daily lifestyle habits including smoking, tooth brushing, and daily physical activity [24,25], the effects of coffee intake on vascular function are still controversial. Several interventional studies have shown acute beneficial effects of coffee intake on vascular function [26,27], while some studies showed no effect [28,29] or harmful effects on vascular function [30,31].

Furthermore, there is little information on the relationship between daily coffee intake and endothelial function, and there is no information on the relationship between daily coffee intake and vascular smooth muscle function. There is also no information on the relationships between amounts of daily coffee intake and vascular function. Therefore, in the present study, we assessed the relationship of daily coffee intake with vascular function using FMD and NID.

## 2. Materials and Methods

### 2.1. Study Subjects

Between April 2016 and August 2021, a total of 462 patients with hypertension who underwent a health checkup at Hiroshima University were enrolled in this cross-sectional study. We excluded the following subjects: subjects being treated with nitrate, subjects with severe chronic heart failure (New York Heart Association level of III or higher), and subjects without information on coffee intake (cups/day). Hypertension was defined as the use of antihypertensive drugs or systolic blood pressure of more than 140 mm Hg or diastolic blood pressure of more than 90 mm Hg measured in a sitting position on at least three occasions. Dyslipidemia was defined according to the third report of the National Cholesterol Education Program [32]. Diabetes mellitus was defined according to the American Diabetes Association recommendation [33]. Smokers were defined as current smokers or former smokers. CVD was defined as coronary heart disease and cerebrovascular disease. Coronary heart disease included angina pectoris, prior myocardial infarction, and unstable angina. Cerebrovascular disease included ischemic stroke, hemorrhagic stroke, and transient ischemic attack. The Ethics Committee of Hiroshima University approved the study protocol. Written informed consent for participation in this study was obtained from all participants.

### 2.2. Study Protocol 

This study was a cross-sectional study. We assessed vascular function in all subjects by using measurements of FMD and NID. The patients fasted overnight and abstained from alcohol, caffeine including coffee, antioxidant vitamins, and smoking for at least 12 h before the study. We divided the subjects into two groups based on information on daily coffee intake: no coffee group and coffee group. We assessed endothelial function and vascular smooth muscle function in each group. Multivariate regression analysis was performed to identify independent variables associated with vascular function. As a post hoc analysis, we fitted a cubic spline curve relationship between the daily amount of coffee intake and vascular function assessed by using FMD and NID. The participants were kept in the supine position in a quiet, dark, air-conditioned room (constant temperature of 22 °C to 25 °C) throughout the study. A 23-gauge polyethylene catheter was inserted into the deep antecubital vein to obtain blood samples. After maintaining the supine position for 30 min, FMD and NID were measured. The observers were blind to the form of examination. 

### 2.3. Measurements of FMD and NID

A high-resolution linear artery transducer was coupled to computer-assisted analysis software (UNEXEF18G, UNEX Co., Nagoya, Japan) that used an automated edge detection system for measurement of the brachial artery diameter [23]. Additional details are available in the Appendix A.

### 2.4. Evaluation of Daily Coffee Intake

Information on daily coffee intake was obtained by using a self-reported questionnaire. The questionnaire consisted of a question on daily coffee intake (yes or no) and the question “How many cups of coffee do you drink each day?”. We defined one cup of coffee as about 200 mL and assessed the amount of coffee intake in the unit of 0.5-cup increments.

### 2.5. Statistical Analysis

Results are presented as means ± SD or medians (interquartile range). Normal distribution was assessed by the Shapiro–Wilk test. All reported probability values were two-sided, and a probability value of <0.05 was considered statistically significant. Categorical values were compared by means of the chi-square test. Leven’s test was used to compare the coffee group and the no-coffee group. Continuous variables were compared by using an unpaired Student’s t-test or Wilcoxson rank-sum test. We categorized subjects into three tertiles based on FMD and NID. The lowest tertile of FMD was 1.6% and the lowest tertile of NID was 8.4%. Therefore, we defined endothelial dysfunction as FMD of <1.6% and vascular smooth muscle dysfunction as NID of <8.4%. Multivariate logistic regression analysis was performed to identify independent variables associated with endothelial dysfunction and vascular smooth muscle dysfunction. Age, sex, body mass index, smokers, presence of dyslipidemia, presence of diabetes mellitus, presence of cardiovascular disease, and systolic blood pressure level were entered into the multivariate logistic regression analysis. The effect of the amount of coffee intake on endothelial function was assessed as a continuous variable by cubic spline curves with the no coffee group as the reference. All data were processed using JMP Pro. Ver 14.0 software (SAS Institute, Cary, NC, USA) and R software version 3.5.1 (R Foundation for Statistical Computing, Vienna, Austria).

## 3. Results

### 3.1. Baseline Characteristics of the Subjects

The baseline clinical characteristics of the 462 patients with hypertension are summarized in Table 1. The 462 subjects included 274 men (59.7%). Among the subjects, 301 (65.2%) had dyslipidemia, 135 (29.2%) had diabetes mellitus, 94 (20.6%) had previous cardiovascular disease and 252 (54.7%) were smokers. The median amount of coffee intake was 1.5 cups/day. The mean FMD value was 3.2 ± 2.9% and the mean NID value was 11.0 ± 5.4%. 

### 3.2. Vascular Function in the No Coffee Group and Coffee Group

The baseline clinical characteristics of the patients with hypertension who did not drink coffee (no coffee group) and patients with hypertension who drank coffee (coffee group) are also summarized in Table 1. There were significant differences in sex, low-density lipoprotein cholesterol, creatinine, prevalence of diabetes mellitus, and prevalence of cardiovascular disease between the two groups. The median coffee intake was two cups per day in the coffee group. The FMD value was significantly lower in the no coffee group than in the coffee group (2.6 ± 2.8% vs. 3.3 ± 2.9%, *p* = 0.04) (Figure 1A). The NID value was significantly lower in the no coffee group than in the coffee group (9.6 ± 5.5% vs. 11.3 ± 5.4%, *p* = 0.02) (Figure 1B). 

Next, we performed multiple logistic analysis to determine whether daily coffee intake was independently associated with endothelial function and vascular smooth muscle function. After adjustments for age, sex, body mass index, systolic blood pressure, presence of dyslipidemia, presence of diabetes mellitus, prevalence of cardiovascular disease, smokers, and systolic blood pressure, the odds ratio for endothelial dysfunction was significantly lower in the coffee group than in the no coffee group (OR: 0.55, 95% CI: 0.32–0.95) (Table 2). After adjustments for age, sex, body mass index, presence of dyslipidemia, presence of diabetes mellitus, presence of cardiovascular disease, and smokers, the odds ratio for vascular smooth muscle dysfunction was significantly lower in the coffee group than in the no coffee group (OR: 0.50, 95% CI: 0.28–0.89) (Table 3).

Finally, we assessed the effects of the amount of coffee intake on endothelial function by cubic spline curves. The cubic spline curves revealed that a daily intake of 0.5 cups to 2.5 cups of coffee had lower odds ratios for endothelial dysfunction and vascular smooth muscle dysfunction (Figure 2A,B). 

## 4. Discussion

In the present study, we demonstrated that both endothelial function and vascular smooth muscle function were more impaired in patients with hypertension who did not drink coffee than in patients with hypertension who drank coffee. An appropriate amount of coffee intake (e.g., less than 2.5 cups/day) might have beneficial effects on endothelial function and vascular smooth muscle function in patients with hypertension.

First, the subjects were divided into two groups of subjects who drank coffee and those who did not drink coffee and vascular function was assessed in each group. Both FMD and NID were significantly smaller in patients with hypertension who did not drink coffee than in patients with hypertension who drank coffee. Next, we assessed the relationship between the amount of daily coffee intake and vascular function in patients with hypertension. Cubic spline curves revealed that the median amount of coffee intake (2 cups per day) decreased the odds ratios for endothelial dysfunction and vascular smooth muscle dysfunction and that excessive coffee intake was harmful to endothelial function and vascular smooth muscle function. Previous studies showed that an excessive amount of caffeine intake elevated blood pressure through the activation of renin activity and activation of the sympathetic nervous system [34,35]. In addition, about 300 mg of caffeine intake per day has been shown to affect blood pressure levels [10]. Furthermore, excessive caffeine intake causes unpleasant symptoms such as depression, headaches, insomnia, and palpitations [36]. In the present study, coffee intake of two cups per day had beneficial effects on endothelial function and vascular smooth muscle function in patients with hypertension. It is likely that coffee intake is a double-edged sword for endothelial function and vascular smooth muscle function depending on its amount.

The mechanisms by which daily coffee intake has beneficial effects on vascular function might be an increase in NO bioavailability caused by compounds in coffee. Coffee contains several compounds including caffeine and chlorogenic acid (CGA), which has an antioxidant property. CGA is one of the polyphenols and is contained in coffee beans. CGA has a protective effect against oxidative stress by scavenging reactive oxygen species and activating adenosine monophosphate-activated protein kinase [37,38]. Our previous study showed that intake of coffee containing a large amount of CGA and a small amount of hydroxy hydroquinone (HHQ), which has a pro-oxidative property, decreased oxidative stress, and improved endothelial function in patients with borderline and stage 1 hypertension. On the other hand, intake of coffee containing a large amount of CGA and a large amount of HHQ had no effect, as did a placebo, on endothelial function in those patients [27]. Caffeine is one of the most popular substances in coffee and also in tea, chocolate, and supplements. Caffeine has several effects on vascular function through various actions including acting as an antagonist of adenosine receptors, increasing intracellular calcium concentration, and inhibiting phosphodiesterase (PDE). Caffeine increases NO production in endothelial cells by binding to the adenosine A1 receptor as an antagonist. In addition, caffeine increases eNOS activity by binding to G-protein-coupled receptors and activating the calcium/calmodulin-dependent protein kinase. It has been shown that caffeine activates cyclic adenosine monophosphate and cyclic guanosine monophosphate through the inhibition of PDE5 [39]. On the other hand, caffeine decreases NO production in endothelial cells by binding to the adenosine A2A receptor as an antagonist. Caffeine per se is also a double-edged sword for vascular function.

This study has potential limitations. First, this study showed the association of daily coffee intake with vascular function. However, this was a single-center, cross-sectional study. Therefore, we cannot define causal relationships between coffee intake with vascular function. Second, only patients with hypertension were recruited in the present study. Assessment of the relationship between vascular function and daily coffee intake in a general population including healthy subjects would enable more specific conclusions to be drawn. Third, the amount of daily coffee intake by Japanese is relatively small. A previous study showed that only about 3% of Japanese adults drank more than five cups of coffee per day [40]. In the present study, the median daily coffee intake was two cups. Therefore, we could not determine the association of a large amount of coffee intake with vascular function. Furthermore, the sample size in the present study was relatively small. Therefore, we cannot clearly indicate that an excessive amount of coffee intake might have negative effects on endothelial function and vascular smooth muscle function in patients with hypertension. Further studies are needed to establish the association of a large amount of coffee intake with vascular function using a large sample size. Fourth, information on daily coffee intake was obtained by using a self-reported questionnaire. However, in previous studies that focused on the relationships of daily coffee intake with variables, information on daily coffee intake was also obtained by using a self-reported questionnaire [6,26,41].

## 5. Conclusions

In conclusion, patients with hypertension who drank two cups of coffee per day had larger FMD and NID than those patients with hypertension who did not drink coffee. Appropriate coffee intake might have beneficial effects on endothelial function and vascular smooth muscle function in patients with hypertension. 

## Figures and Tables

**Figure 1 nutrients-14-02719-f001:**
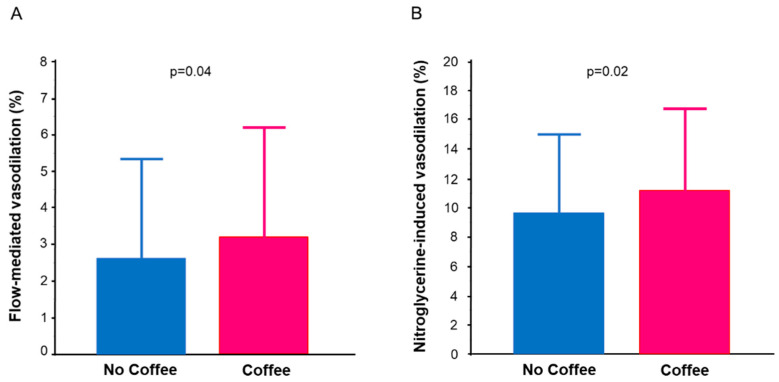
Bar graphs show flow-mediated vasodilation (**A**) and nitroglycerine-induced vasodilation (**B**) in patients with hypertension who did not drink coffee and patients with hypertension who drank coffee.

**Figure 2 nutrients-14-02719-f002:**
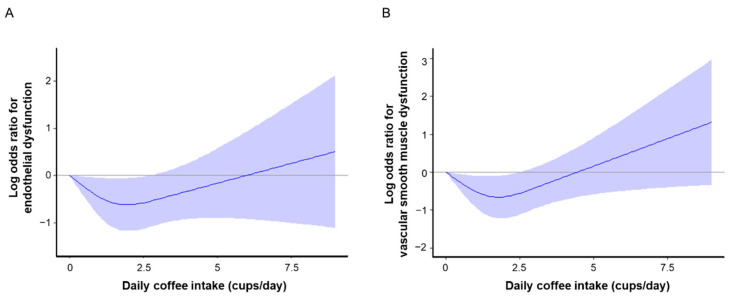
Cubic splines of the relationships of amount of daily coffee intake with log odds ratios for endothelial dysfunction (**A**) and vascular smooth muscle function (**B**). Endothelial dysfunction was defined as flow-mediated vasodilation of less than 1.6% and vascular smooth muscle dysfunction was defined as nitroglycerine-induced vasodilation of less than 8.4%. Vertical lines show 95% confidence intervals.

**Table 1 nutrients-14-02719-t001:** Clinical Characteristics of Subjects in Groups According to Daily Coffee Intake.

Variables	Total (*n* = 462)	No Coffee (*n* = 71)	Coffee (*n* = 391)	*p* Value
Age, year	65 ± 13	67 ± 13	64 ± 13	0.08
Men, n (%)	274 (59.7)	51 (71.8)	223 (57.5)	0.02
Body mass index, kg/m^2^	24.4 ± 3.8	24.5 ± 3.8	24.4 ± 3.7	0.84
Heart rate, bpm	69 ± 11	70 ± 14	69 ± 11	0.49
Systolic blood pressure, mmHg	130 ± 17	133 ± 20	129 ± 16	0.10
Diastolic blood pressure, mmHg	79 ± 12	81 ± 12	78 ± 12	0.10
Total cholesterol, mmol/L	4.91 ± 0.93	4.73 ± 0.88	4.94 ± 0.93	0.12
Triglycerides, mmol/L	1.28 (0.93, 1.81)	1.31 (0.86, 1.87)	1.26 (0.94, 1.80)	0.63
High-density lipoprotein cholesterol, mmol/L	1.55 ± 0.41	1.55 ± 0.39	1.55 ± 0.44	0.97
Low-density lipoprotein cholesterol, mmol/L	2.79 ± 0.78	2.61 ± 0.75	2.84 ± 0.78	0.04
Creatinine, mmol/L	71.6 (61.0, 85.8)	84.9 (64.5, 97.2)	70.7 (60.1, 84.6)	<0.01
Glucose, mmol/L	5.94 ± 1.28	5.88 ± 1.33	5.94 ± 1.28	0.73
Hemoglobin A1c, %	5.9 ± 0.9	5.8 ± 1.3	5.9 ± 0.8	0.59
Medical history, n (%)				
Dyslipidemia	301 (65.2)	48 (67.6)	253 (64.7)	0.64
Diabetes mellitus	135 (29.2)	14 (19.7)	121 (31.0)	0.048
Cardiovascular disease, n (%)	94 (20.6)	23 (32.4)	71 (18.4)	<0.01
Smoker, n (%)	252 (54.7)	39 (54.9)	213 (54.6)	0.96
Medication, n (%)				
Antihypertensive drugs	421 (91.9)	65 (92.9)	356 (91.8)	0.75
Lipid-lowering drugs	190 (62.5)	30 (62.5)	160 (62.5)	1.00
Anti-diabetic drugs	130 (28.1)	14 (19.7)	116 (29.7)	0.09

**Table 2 nutrients-14-02719-t002:** Multivariate Analysis of Endothelial Dysfunction in No Coffee and Coffee Groups.

	Odds Ratio (95% Confidence Interval); *p* Value				
Coffee	Unadjusted	Model 1	Model 2	Model 3	Model 4
No coffee	1 (reference)	1 (reference)	1 (reference)	1 (reference)	1 (reference)
Coffee	0.57 (0.34–0.95); 0.03	0.58 (0.34–0.99); 0.04	0.56 (0.33–0.96); 0.04	0.55 (0.32–0.95); 0.03	0.55 (0.32–0.95); 0.03

Model 1: adjusted for age, sex, and body mass index. Model 2: adjusted for age, sex, body mass index, dyslipidemia, diabetes mellitus, and smokers. Model 3: adjusted for age, sex, body mass index, dyslipidemia, diabetes mellitus, smoker, cardiovascular disease. Model 4; adjusted for age, sex, body mass index, dyslipidemia, diabetes mellitus, smokers, cardiovascular disease, and systolic blood pressure. Endothelial dysfunction was defined as FMD of less than 1.6%.

**Table 3 nutrients-14-02719-t003:** Multivariate Analysis of Vascular Smooth Muscle Dysfunction in No Coffee and Coffee Groups.

	Odds ratio (95% Confidence Interval); *p* Value				
Coffee	Unadjusted	Model 1	Model 2	Model 3	Model 4
No coffee	1 (reference)	1 (reference)	1 (reference)	1 (reference)	1 (reference)
Coffee	0.51 (0.30–0.86); 0.01	0.49 (0.28–0.84); 0.01	0.45 (0.25–0.78); <0.01	0.49 (0.27–0.87); 0.02	0.50 (0.28–0.89); 0.02

Model 1: adjusted for age, sex, and body mass index. Model 2: adjusted for age, sex, body mass index, dyslipidemia, diabetes mellitus, and smokers. Model 3: adjusted for age, sex, body mass index, dyslipidemia, diabetes mellitus, smoker, cardiovascular disease. Model 4; adjusted for age, sex, body mass index, dyslipidemia, diabetes mellitus, smokers, cardiovascular disease, and systolic blood pressure. Vascular smooth muscle dysfunction was defined as NID of less than 8.4%.

## Data Availability

The data presented in this study are available on request from the corresponding author. The data are not publicly available due to institutional policies requiring a data-sharing agreement.

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
