# Peer review of "Relationship of Daily Coffee Intake with Vascular Function in Patients with Hypertension"

_nutrients, 2022, doi:10.3390/nu14132719_

Round 1

Reviewer 1 Report

This is a novel association study of the relationship between coffee intake and vascular function in Japanese patients with hypertension.  The study was conducted by a distinguished group of investigators at Hiroshima University.  The corresponding author is an internationally recognized expert in cardiovascular research and in the assessment of endothelial and vascular smooth muscle function. The relationship of coffee to health is a topic of considerable popular interest and will attract readers.    

The authors found that endothelial function and vascular smooth muscle function are greater in subjects consuming 2 cups of coffee per day than in those consuming no coffee (after adjustments for multiple confounding factors).    I have several comments to improve the manuscript that should be straightforward to address with some minor revisions to the manuscript.

1. In the abstract, it is stated that ” After adjustment for confounding factors, the odds ratio of  the lowest tertile of flow-mediated vasodilation (OR: 0.56, 95% CI: 0.33-0.97) and lowest tertile of nitroglycerine-induced vasodilation (OR: 0.43, 95% CI: 0.24-0.76) were significantly lower in the coffee group than in the no coffee group.”   I found this a little difficult to understand.   It is easier to understand the abstract  if you say “After adjustment for confounding factors, the odds ratio for  endothelial dysfunction (OR: 0.56, 95% CI: 0.33-0.97) and the odds ratio for vascular smooth muscle dysfunction (OR: 0.43, 95% CI: 0.24-0.76) were significantly lower in the coffee group relative to the no coffee group.”  

2. In the introduction, it is stated that  “Several studies have shown that coffee intake and even excessive intake decreased CVD events,[9,10] while other studies showed that even excessive coffee intake also decreased CVD events.[11,12]”   This sentence seems to have some redundancy.  I suggest you say  that  “Some studies have shown that coffee intake or even excessive intake can be associated with decreased CVD events,[9,10, 11, 12].”

3.  In the last paragraph of the introduction, it is stated that “Furthermore, there is little information on the relationship between daily coffee intake and vascular function, and there is no information on the relationship between daily coffee intake and vascular smooth muscle function.”     In the first part of this sentence,  you refer to “vascular function” but it is not clear what type of vascular function you are referring to.  Do you mean to say the following? “Furthermore, there is little information on the relationship between daily coffee intake and endothelial function, and there is no information on the relationship between daily coffee intake and vascular smooth muscle function.”  

4. Please clarify the methods section. Were the patients fasting or was coffee withheld on the day of the measurements?   When were the measurements made in relation to coffee consumption on the  day of the measurements ?  Is it possible some people were tested shortly after consuming coffee that day? 

5.  I have some questions regarding the results section lines 153 – 170 where you describe differences between the coffee group and no coffee group and also describe adjustments made for confounding factors.   On lines 155-157, it is stated that “There were significant differences in sex, low density lipoprotein cholesterol, creatinine, prevalence of diabetes mellitus, prevalence of  current smokers, and prevalence of subjects who used anti-diabetic drugs between the  two groups.”   According to the table, it also appears that there was a difference in the prevalence of cardiovascular disease between the two groups.   Is there some reason you did not mention this difference in CVD prevalence when describing the differences between the two groups?  Please clarify in the manuscript.

On lines 164 – 165, you mention the adjustments made for age, sex, and other potential confounders.   However, you do not mention adjustment made for differences in cardiovascular disease prevalence. Is there some reason you did not adjust for this ?  Please clarify in the manuscript.    Is it possible that coffee drinking influences the CVD prevalence and that the lower vascular function in the no coffee group is secondary to more CV disease in that group ?

6. The sentences on lines 190 – 192 appear to be instructions to the authors.  Please remove the following sentences from the manuscript:  “This section may be divided by subheadings. It should provide a concise and precise description of the experimental results, their interpretation, as well as the experimental conclusions that can be drawn.”

7. The first sentence of the discussion is long.  Please  break it up into two  sentences.  In addition, the readers may not remember how "excessive" is defined.   Please redefine the meaning of "excessive' here.  Alternatively, do not use the word excessive and  simply refer to “drinking more than XX cups/day.”  

Author Response

We would like to thank the reviewer for the helpful comments and hope that we have now produced a more balanced and better account of our work.

  1. In the abstract, it is stated that ”After adjustment for confounding factors, the odds ratio of the lowest tertile of flow-mediated vasodilation (OR: 0.56, 95% CI: 0.33-0.97) and lowest tertile of nitroglycerine-induced vasodilation (OR: 0.43, 95% CI: 0.24-0.76) were significantly lower in the coffee group than in the no coffee group.” I found this a little difficult to understand. It is easier to understand the abstract if you say “After adjustment for confounding factors, the odds ratio for endothelial dysfunction (OR: 0.56, 95% CI: 0.33-0.97) and the odds ratio for vascular smooth muscle dysfunction (OR: 0.43, 95% CI: 0.24-0.76) were significantly lower in the coffee group relative to the no coffee group.”

Response: In accordance with the reviewer’s appropriate suggestion, the sentence “After adjustment for confounding factors, the odds ratio of the lowest tertile of flow-mediated vasodilation (OR: 0.56, 95% CI: 0.33-0.97) and lowest tertile of nitroglycerine-induced vasodilation (OR: 0.43, 95% CI: 0.24-0.76) were significantly lower in the coffee group than in the no coffee group.” has been changed to “After adjustment for confounding factors, the odds ratio for endothelial dysfunction (OR: 0.56, 95% CI: 0.33-0.97) and the odds ratio for vascular smooth muscle dysfunction (OR: 0.43, 95% CI: 0.24-0.76) were significantly lower in the coffee group than in the no coffee group.”

  1. In the introduction, it is stated that “Several studies have shown that coffee intake and even excessive intake decreased CVD events,[9,10] while other studies showed that even excessive coffee intake also decreased CVD events.[11,12]” This sentence seems to have some redundancy. I suggest you say that “Some studies have shown that coffee intake or even excessive intake can be associated with decreased CVD events,[9,10, 11, 12].”

Response: In accordance with the reviewer’s appropriate suggestion, the sentence “Several studies have shown that coffee intake and even excessive intake decreased CVD events,[9,10] while other studies showed that even excessive coffee intake also decreased CVD events.[11,12]“ has been changed to “Some studies have shown that coffee intake or even excessive intake can be associated with decreased CVD events,[9,10, 11, 12].

  1. In the last paragraph of the introduction, it is stated that “Furthermore, there is little information on the relationship between daily coffee intake and vascular function, and there is no information on the relationship between daily coffee intake and vascular smooth muscle function.” In the first part of this sentence, you refer to “vascular function” but it is not clear what type of vascular function you are referring to. Do you mean to say the following? “Furthermore, there is little information on the relationship between daily coffee intake and endothelial function, and there is no information on the relationship between daily coffee intake and vascular smooth muscle function.”

Response: We apologize for the mistake in line 80. In accordance with the reviewer’s appropriate suggestion, the term “vascular function” has been corrected to “endothelial function.”

  1. Please clarify the methods section. Were the patients fasting or was coffee withheld on the day of the measurements? When were the measurements made in relation to coffee consumption on the day of the measurements? Is it possible some people were tested shortly after consuming coffee that day?

Response: In accordance with the reviewer’s appropriate suggestion, we have added more detailed information on measurement of FMD to the Methods section (lines 105-107). The patients fasted overnight and abstained from alcohol, caffeine including coffee, antioxidant vitamins and smoking for at least 12 hours before the study. Therefore, in the present study, we assessed the chronic effects of daily coffee intake on vascular function.

  1. I have some questions regarding the results section lines 153 – 170 where you describe differences between the coffee group and no coffee group and also describe adjustments made for confounding factors. On lines 155-157, it is stated that “There were significant differences in sex, low density lipoprotein cholesterol, creatinine, prevalence of diabetes mellitus, prevalence of current smokers, and prevalence of subjects who used anti-diabetic drugs between the two groups.” According to the table, it also appears that there was a difference in the prevalence of cardiovascular disease between the two groups. Is there some reason you did not mention this difference in CVD prevalence when describing the differences between the two groups? Please clarify in the manuscript.

Response: In accordance with the reviewer’s appropriate suggestion, we have added information on the prevalence of cardiovascular disease in the coffee group and no coffee group to the Results section (line 159).

On lines 164 – 165, you mention the adjustments made for age, sex, and other potential confounders. However, you do not mention adjustment made for differences in cardiovascular disease prevalence. Is there some reason you did not adjust for this?  Please clarify in the manuscript. Is it possible that coffee drinking influences the CVD prevalence and that the lower vascular function in the no coffee group is secondary to more CV disease in that group?

Response: In accordance with the reviewer’s appropriate suggestion, we performed the multivariate analysis including the factor of cardiovascular disease prevalence. After adjustment for age, sex, body mass index, dyslipidemia, diabetes mellitus, smokers, presence of cardiovascular disease, and systolic blood pressure, the odds ratio for endothelial dysfunction (OR: 0.55, 95% CI: 0.32-0.95) and the odds ratio for vascular smooth muscle dysfunction (OR: 0.50, 95% CI: 0.28-0.89) were significantly lower in the coffee group than in the no coffee group. These results have been added to the Results section (lines 174, 176, 178, and 180) and in Tables 2 and 3.

In accordance with the reviewer’s appropriate suggestion, we reanalyzed the results by incorporating CVD prevalence into the multivariate analysis and the results did not change. Therefore, we reject the possibility that coffee drinking influences the CVD prevalence and that the decline in vascular function in the no coffee group is secondary to more CV disease in that group.

  1. The sentences on lines 190 – 192 appear to be instructions to the authors. Please remove the following sentences from the manuscript: “This section may be divided by subheadings. It should provide a concise and precise description of the experimental results, their interpretation, as well as the experimental conclusions that can be drawn.”

Response: In accordance with the reviewer’s appropriate suggestion, we removed sentences “This section may be divided by subheadings. It should provide a concise and precise description of the experimental results, their interpretation, as well as the experimental conclusions that can be drawn.”

  1. The first sentence of the discussion is long. Please break it up into two sentences. In addition, the readers may not remember how "excessive" is defined. Please redefine the meaning of "excessive' here. Alternatively, do not use the word excessive and simply refer to “drinking more than XX cups/day.”

Response: In accordance with the reviewer’s appropriate suggestion, the first sentence in the discussion section has been rewritten as two senteces. (lines 205-210)

In the present study, we did not define an excessive amount of coffee. We assessed the cubic splines of the relationships between coffee intake and odds ratios for endothelial dysfunction assessed by FMD and vascular smooth muscle dysfunction assessed by NID (Figure 2A and 2B). The cubic spline curves revealed that daily intake of 0.5 cups to 2.5 cups of coffee has lower odds ratios for endothelial dysfunction and vascular smooth muscle dysfunction. Therefore, we have rewritten the sentense “ an excessive amount of coffee intake might not have beneficial effects on vascular function in patients with hypertension” as “an appropriate amount of coffee intake (e.g., less than 2.5 cups/day) might have beneficial effects on endothelial function and vascular smooth muscle function in patients with hypertension” (lines 207-210).

Reviewer 2 Report

Dear Editor,

I carefully read the manuscript by Yamaji et al.

My comments and suggestions for the authors are the following:

 - English language needs to be carefully improved.

 - The authors should specify how the normal distribution of the parameters was assessed.

 - The authors should specify if they performed the Levene's test before the Student's test.

 - Lines 136-138: How did the authors choose to include these variables in the logistic analysis? Did they perform a univariate analysis a priori?

 - Lines 106-107: How was the coffee intake assessed? Did the authors use a standardized questionnaire (e.g. the 7-day FFQ)?

 - Table 1: "Hypertension" should not be included among the variables since it is the most important inclusion criterion for the study.

 - The limitations of the analysis should be more detailed in the discussion.

Author Response

MS ID#: nutrients-1764771R1                   Reviewer #2

We would like to thank the reviewer for the helpful comments and hope that we have now produced a more balanced and better account of our work.

  1. English language needs to be carefully improved.

Response: We consulted an English native speaker to check the English language in the manuscript.

  1. The authors should specify how the normal distribution of the parameters was assessed.

Response: In accordance with the reviewer’s appropriate suggestion, we performed the Shapiro-Wilk test to assess the normal distribution of parameters. Triglycerides and creatine did not have normal distributions. Therefore, we reassessed these parameters by the Wilcoxson rank sum test. These comments have been incorporated into the Methods section (lines 132-133, and 137) and Table 1.

3.The authors should specify if they performed the Levene's test before the Student's test.

Response: In accordance with the reviewer’s appropriate suggestion, we performed Levene’s test before Student’s test.

4 Lines 136-138: How did the authors choose to include these variables in the logistic analysis? Did they perform a univariate analysis a priori?

Response: In accordance with the reviewer’s appropriate suggestion, we explain why we selected variables in the logistic analysis. We showed that both FMD and NID decreased in relation to cumulative cardiovascular risk factors and significantly correlated with cardiovascular risk factors (Maruhashi, et al. Heart. 2013;99:1837-1842. Maruhashi, et al. Hypertension Res. 2020;43:914-921. Matsui, et al. Sci Rep 2017;7:8422. Yamaji, et al. BMJ Open Diab Res Care.2020;0:e001610. Maruhashi, et al. Arterioscler Thromb Vasc Biol. 2013;33:1401-1408.).

Therefore, sex, body mass index, smokers, presence of dyslipidemia, presence of diabetes mellitus, presence of cardiovascular disease, and systolic blood pressure level were entered into the multivariate logistic regression analysis.

  1. - Lines 106-107: How was the coffee intake assessed? Did the authors use a standardized questionnaire (e.g. the 7-day FFQ)?

Response: All of the information on coffee intake was obtained by a questionnaire as stated in the previous version (lines 123-126). Participants reported daily coffee intake in a day and in a week. We defined 1 cup of coffee as about 200 mL and assessed the amount of coffee intake in the unit of 0.5-cup increments.

  1. - Table 1: "Hypertension" should not be included among the variables since it is the most important inclusion criterion for the study.

Response: In accordance with the reviewer’s appropriate suggestion, we have removed the word “Hypertension” in Table 1.

  1. - The limitations of the analysis should be more detailed in the discussion.

Response: In accordance with the reviewer’s appropriate suggestion, we have added more detailed limitations of the analysis to the discussion section. First, the sample size in the present study was relatively small. Therefore, we cannot clearly indicate that excessive amount of coffee intake might have negative effects on endothelial function and vascular smooth muscle function in patients with hypertension. Further studies are needed to establish the association of a large amount of coffee intake with vascular function using a large sample size. These comments have been incorporated into the Discussion section (lines 212-215).

Round 2

Reviewer 2 Report

Dear Editor,

I carefully read the revised version of the manuscript that is significantly improved in comparison with the original version. I have no more comments on this interesting paper.